# Clinical Potential of Hydrogen Sulfide in Peripheral Arterial Disease

**DOI:** 10.3390/ijms24129955

**Published:** 2023-06-09

**Authors:** Clémence Bechelli, Diane Macabrey, Sebastien Deglise, Florent Allagnat

**Affiliations:** Department of Vascular Surgery, Lausanne University Hospital, 1005 Lausanne, Switzerland; clemence.bechelli@unil.ch (C.B.); sebastien.deglise@chuv.ch (S.D.)

**Keywords:** peripheral artery disease, PAD, intimal hyperplasia, hydrogen sulfide, H_2_S, atherosclerosis, inflammation, calcification

## Abstract

Peripheral artery disease (PAD) affects more than 230 million people worldwide. PAD patients suffer from reduced quality of life and are at increased risk of vascular complications and all-cause mortality. Despite its prevalence, impact on quality of life and poor long-term clinical outcomes, PAD remains underdiagnosed and undertreated compared to myocardial infarction and stroke. PAD is due to a combination of macrovascular atherosclerosis and calcification, combined with microvascular rarefaction, leading to chronic peripheral ischemia. Novel therapies are needed to address the increasing incidence of PAD and its difficult long-term pharmacological and surgical management. The cysteine-derived gasotransmitter hydrogen sulfide (H_2_S) has interesting vasorelaxant, cytoprotective, antioxidant and anti-inflammatory properties. In this review, we describe the current understanding of PAD pathophysiology and the remarkable benefits of H_2_S against atherosclerosis, inflammation, vascular calcification, and other vasculo-protective effects.

## 1. Introduction

Peripheral artery disease (PAD), defined as “all arterial diseases other than coronary arteries and aorta”, affects more than 230 million people worldwide [1,2].

PAD is primarily due to the development of atherosclerotic plaques, leading to progressive narrowing of the vessel lumen. Limb symptoms include leg pain, cramps, fatigue, and muscle weakness during physical activity. At rest, blood flow remains sufficient to meet basal oxygen requirements and patients are free of symptoms. However, during exercise, the increased oxygen supply to the lower limb is impaired, leading to moderate ischemia, which the patient experiences as cramping pain. The patient usually stops walking until the pain subsides. Alternating cycles of walking and resting, known as intermittent claudication (IC), is the cardinal clinical manifestation of PAD [3]. Patients with IC have a reduced walking distance, leading to an inability to perform daily activities and a reduced quality of life [1,4,5]. However, IC may be present in only 10–35% of patients, whereas 40–50% of PAD patients have a wide range of atypical leg symptoms, and 20–50% of patients are asymptomatic [4,5,6]. The femoral and popliteal arteries are the most common sites of atherosclerotic disease in patients with PAD. Approximately 80–90% of patients with symptomatic PAD have some combination of femoropopliteal occlusive disease [4,5,7].

In late-stage PAD, ischemia worsens as the arteries become completely occluded, leading to chronic limb-threatening ischemia (CLTI). CLTI is characterized by resting muscle pain, ulceration, and gangrene, and a significant reduction in quality of life. In addition, PAD and CLTI patients are at increased risk of developing vascular occlusive disease and all-cause mortality, as atherosclerosis usually develops throughout the vasculature. Notably, the 1-year incidence of all major cardiovascular events is 30% higher in patients with PAD than in those with coronary or cerebral artery disease [8]. Without surgical revascularization, 25% of CLTI patients die within one year of initial diagnosis and 40% of CLTI patients undergo limb amputation within three years [9,10]. Venous bypass surgery and endovascular approaches such as angioplasty with or without stenting and endarterectomy are the main treatment for CLTI. The disease presentation and the patient’s general health and comorbidities determine the choice between open surgery and endovascular approaches.

Acute limb ischemia (ALI) is another severe manifestation of PAD, defined by sudden, severe hypoperfusion of the limb, usually due to thromboembolism. Symptoms may include pain, pallor, pulselessness, poikilothermia, paresthesias and paralysis, with loss of sensation and motor function in severe cases. Although ALI can occur in the absence of significant peripheral atherosclerosis due to distant plaque rupture, it is common in the setting of PAD.

Despite its prevalence, impact on quality of life, and devastating long-term clinical outcomes, PAD remains underdiagnosed and undertreated compared with other atherosclerotic diseases such as myocardial infarction and stroke [2,11].

## 2. Current Management of PAD and CLTI

The main risk factors for the development of PAD are age, smoking, and diabetes. Hyperlipidemia and hypertension are also risk factors for PAD, although the predictive value of these parameters does not appear to be as strong as for the primary risk factors. The presentation of PAD varies considerably and includes four categories: asymptomatic, claudication, critical limb ischemia, and ALI. PAD patients are classified according to the Fontaine or Rutherford classification systems.
**Fontaine**
Stage I—No symptomsStage II—Intermittent claudication subdivided into:Stage IIa—Without pain on resting, but with claudication at a distance of greater than 650 feet (200 m)Stage IIb—Without pain on resting, but with a claudication distance of less than 650 feet (200 m)Stage III—Nocturnal and/or resting painStage IV—Necrosis (death of tissue) and/or gangrene in the limb
**Rutherford**
Stage 0—AsymptomaticStage 1—Mild claudicationStage 2—Moderate claudicationStage 3—Severe claudicationStage 4—Rest painStage 5—Minor tissue loss with ischemic nonhealing ulcer or focal gangrene with diffuse pedal ischemiaStage 6—Major tissue loss—Extending above transmetatarsal level, functional foot no longer salvageable

Asymptomatic PAD patients with evidence of atherosclerosis who do not have typical claudication symptoms (Fontaine I or Rutherford 0) are offered risk reduction strategies to decrease cardiovascular risk factors depending on symptom severity, lipid levels, and the presence of comorbidities such as diabetes, smoking and hypertension. Thus, current guidelines for the management of PAD are preventive strategies such as diet and lifestyle modification, including supervised exercise, smoking cessation and pharmacotherapy tailored to individual risk factors [1,8,12,13,14,15]. All patients with PAD should receive statin medication. Antihypertensive therapy should be administered to hypertensive patients to reduce the risk of myocardial infarction (MI), stroke, heart failure, and cardiovascular death. Antiplatelet therapy with aspirin or clopidogrel alone may be considered in asymptomatic patients, and should always be administered to symptomatic PAD patients. After assessment of bleeding risk, further anti-coagulant therapies (Rivaroxaban) may be considered for symptomatic PAD patients as they significantly reduce the risk of stroke, myocardial infarction, and ALI [1,12,13,16,17].

For patients with lifestyle-limiting claudication or CLTI (Fontaine IIb—IV; Rutherford 4–6), who are poor responders to medical and/or exercise therapy, surgical revascularization remains the only option when possible. Venous bypass surgery and endovascular approaches such as angioplasty, stenting and atherectomy are the main methods. The choice between open surgery and endovascular approaches depends on the presentation of the disease and the patient’s general health and comorbidities. Whenever possible, autogenous vein is the conduit of choice for open revascularization so that bypass surgery is limited to patients with “good” veins [7,18]. All patients with CLTI should be given antithrombotic and lipid-lowering therapies, as well as counseling on smoking cessation, diet, exercise, and preventive foot care. Additional antihypertensive, and glycemic control therapies should be given appropriately [1,12,13].

Without surgical revascularization, 25% of CLTI patients die within one year of initial diagnosis and 40% of CLTI patients undergo limb amputation within three years [9,10]. Up to 25% of CLTI patients are ineligible for revascularization and amputation is often the only option [19]. When possible, surgery may be suboptimal for symptom relief, and 20% of PAD patients have “failed revascularization”. Furthermore, PAD patients, especially those with CLTI, carry a high risk of post-op complications, including ALI, often leading to limb loss, disability, and death [13,20]. Even if the procedure is technically successful, residual microvascular disease remains and the outcomes after amputation stay poor [13,21].

## 3. Etiology of PAD

Atherosclerosis in lower limb arteries is the main cause of PAD [22], but emerging evidence suggests that medial calcification also contributes to the disease, especially in lower limb PAD. Microvascular disease is also emerging as a potential contributor to the progression of PAD and a clinically relevant sign of PAD severity.

### 3.1. Atherosclerosis

Atherosclerosis is a chronic inflammatory disease characterized by the accumulation of fatty cholesterol streaks in arterial trees. Several pathophysiological processes are involved in this disease, including endothelial cell (EC) dysfunction, inflammation, lipid accumulation, and vascular smooth muscle cell (VSMC) proliferation and migration (reviewed in detail in [23]).

The disease is initiated by EC dysfunction. Located at the interface between the blood and the vessel wall, EC maintain a non-thrombogenic surface. In arteries, high shear stress and laminar blood flow maintain EC function and secretion of anti-thrombotic and vasodilator agents, mainly nitric oxide (NO) and prostacyclins [24]. Disturbed arterial flow patterns observed at bifurcations and curved sections of arteries create regions of low shear stress that induce EC dysfunction or “endothelial activation”. These weak points in the vasculature are the sites of primary occlusion by atherosclerotic plaques. Endothelial dysfunction or injury results in reduced production of NO and hydrogen sulfide (H_2_S), two gasotransmitters that maintain healthy vascular function. Impaired EC function promotes vasoconstriction, platelet aggregation and the accumulation of oxidized low-density lipoproteins (LDL) in the vessel wall. Monocytes attracted to the inflamed vessel wall differentiate into macrophages, which engulf large amounts of LDL particles and become foam cells to form the fatty streaks typical of early atherosclerotic lesions. Foam cells undergo apoptosis and form a lipid core within the vessel wall, exacerbating inflammation. The VSMC composing the media layer of vessels are highly plastic. Upon chronic inflammation, VSMC switch to a “synthetic” phenotype, characterized by a loss of contractile markers. Recent lineage-tracing studies revealed that VSMC dedifferentiate into intermediate multipotent cell type, often referred to as mesenchymal stem cells (MSC). These cells may give rise to adipocytes, myofibroblasts, macrophage-like cells and fibro/osteochondrogenic cells [25,26,27,28]. Of note, VSMC-derived macrophages perform nonprofessional phagocytosis and contribute to the population of foam cells in atherosclerotic plaques [29,30]. Altogether, proliferating immune cells and reprogrammed VSMC promote matrix remodeling and the development of a fibrous cap overlying the lipid core.

Overall, atherosclerosis is driven by dyslipidemia and vascular chronic inflammation [28,31]. Macrophages are the primary immune cells involved in atherosclerosis, but over the years evidence has accumulated of a coordinated inflammatory immune response involving T- and B-lymphocytes in the progression of atherosclerotic plaques [28]. It should also be noted that all the cell types found in atheromatous plaques can secrete pro-inflammatory cytokines, such as interleukin-1 (IL-1) and tumor necrosis factors alpha (TNFα) and chemokine monocyte chemoattractant protein-1 (MCP-1/CCL2). Activated T-helper 1 (Th1) lymphocytes produce interferon gamma (IFNγ), which promotes phagocytosis and formation of foam cells. B2 lymphocytes also secrete mediators that can aggravate atherogenesis. In contrast, other immune cells including M2 macrophages, B1 lymphocytes and Th2 lymphocytes can produce anti-inflammatory mediators to alleviate inflammation [28,31]. In addition, activated EC secrete lipid-derived pro-inflammatory molecules called eicosanoids, including prostaglandins, leukotrienes, and thromboxanes, which also play a major role in the pathophysiology of atherosclerosis [32,33].

Despite decades of research and although dyslipidemia and inflammation are known to be the major pathophysiological features leading to atherosclerosis, the exact pathways and mechanisms remain to be elucidated.

### 3.2. Vascular Medial Calcification

PAD is commonly described as an atherosclerotic disease. However, for lower limb artery disease, recent clinical data suggest that we underestimated the role of medial arterial calcification in PAD (recently reviewed in detail in [34,35]). Thus, the etiology of PAD, particularly in the arteries below the knee, may differ from that of the coronary and femoral arteries.

Two types of vascular calcification exist, intimal calcification (VIC) and medial calcification (VMC), also referred to as medial arterial calcification (MAC) [34,35]. VIC is a common feature of advanced atherosclerotic lesions and a risk factor for rupture. In contrast, VMC/MAC develops independently of atherosclerosis, but is a common feature of arterial disease associated with aging [36]. It is found in up to 40% of patients with advanced chronic kidney disease [37,38,39,40], and histological studies show that up to 70% of occluded arteries below the knee feature VMC and intimal thickening, but no atherosclerotis [41]. In their recent study, Jadidi et al. used machine learning to identify age, creatinine, body mass index, coronary artery disease and hypertension as the strongest predictors of calcification. They further confirmed that distal vessel segments (iliofemoral vs. aortic) calcify first. In this study of an American cohort, they estimated that up to 80% of people had VMC by the age of 40 [36].

VMC is characterized by the accumulation of calcium (Ca^2+^) phosphate and the formation of hydroxyapatite crystals, leading to hardening of the medial layer [38]. It is particularly prevalent in patients with chronic kidney disease, especially diabetic patients, due to impaired phosphate homeostasis [35,39,40]. Different stages/severities of arterial calcification have been described by histopathologists, ranging from punctate to nodular calcification, and finally bone formation [34].

VIC in atherosclerosis lesion is well characterized. It is due to ectopic vascular osteogenesis via phenotypic reprogramming of contractile medial VSMC into synthetic mesenchymal VSMC, which then differentiate into osteochondrogenic VSMC, leading to bone formation [35]. VMC in lower limb arteries has not been so well studied. The presence of osteogenesis vs. hydroxyapatite deposition and their respective contribution to VMC in PAD and CLTI patients remain unknown, and may differ depending on the vascular bed [38,39,40]. VMC increases the risk of complications during vascular interventions and worsens their outcomes [34,35,42]. Further work is required to define the process underlying medial calcification in the absence of atherosclerosis, evaluate its impact on PAD and CLTI, and eventually target it for treatment.

### 3.3. Microvascular Dysfunction

PAD is usually recognized as a macrovascular disease. However, several recent studies indicate that artery occlusion in PAD is often accompanied by microvascular disease. Microvascular dysfunction (MVD) refers to the impairment of capillary function and number. Usually, peripheral microvascular endothelial function is evaluated using laser speckle contrast imaging, which allows assessment of cutaneous microcirculation. The incidence of MVD is particularly high in diabetic patients. Thus, 20 to 30% of PAD patients, and up to 70% of CLTI patients have diabetes [10]. Of note, diabetic patients have a five-fold increased risk of developing CLTI, and diabetic CLTI patients have up to five-fold more incidence of adverse outcomes and amputations [9,10,43]. Given the strong association between diabetes complications and MVD, clinical studies also tend to define MVD as the presence of nephropathy, retinopathy, or neuropathy. Clinical studies revealed a strong association between MVD and risk of heart failure in diabetic patients, independent of traditional heart failure risk factors including coronary artery disease [44,45,46]. MVD is also a common phenomenon in PAD patients, which feature impaired cutaneous microcirculation throughout the progression of PAD, often leading to reduced capillary density in CLTI patients. In PAD patients, MVD can contribute to the progression of the disease and the development of complications such as ischemic pain, tissue hypoxia, and impaired wound healing [10]. A recent study also found a positive correlation between microvascular endothelial function and impaired cognitive performance in PAD patients [47]. MVD can also worsen the outcome of surgical procedures as it reduces the ability of the blood vessels to respond to the increased blood flow after revascularization, which impairs healing, leading to a higher risk of complications.

Additionally, recent studies suggest that MVD may be used to assess PAD severity. In a recent meta-analysis, the Chronic Kidney Disease Prognosis discovered that albuminuria, a marker of nephropathy, strongly correlates with the incidence of amputation [48]. This study advocates that even at mild-to-moderate stages, chronic kidney disease and MVD may be a major risk factor for PAD. In a similar study, a stronger association was found between retinopathy and the incidence of PAD/CLTI, than between coronary heart disease or stroke and PAD/CLTI [49].

Mechanistically, MVD is not due to the formation of atherosclerosis plaque and/or occlusion of vessels. MVD is due to EC apoptosis and progressive loss of capillaries, which plays a major role in the development and progression of diabetic complications (diabetic retinopathy, nephropathy, and neuropathy). Patients with familial hypercholesterolemia also feature impaired endothelial-dependent vasodilatation [50].

Overall, MVD contributes to PAD, but is seldom considered in diagnostic and therapeutic approaches. There is currently no specific therapy for MVD. However, the good news is that current PAD therapeutic strategies focused on optimizing risk factors (management of diabetes, and hypercholesterolemia), and lifestyle modifications (physical exercise, smoking cessation, and weight loss), improve vascular fitness, including microvascular function. For instance, several clinical studies demonstrated that exercise promotes microvascular function in disease states [51,52,53,54,55]. Although solid evidence is still lacking, statins may also provide benefits to endothelial function and against MVD [56,57]. Pre-clinical studies also showed that anti-diabetic therapies, metformin especially, may preserve/restore endothelium function [58,59,60,61]. Understanding the mechanisms underlying MVD in PAD patients and finding new treatments and therapeutics targeting MVD specifically may help reduce symptoms and improve quality of life.

Overall, PAD is due to a combination of macrovascular atherosclerosis and calcification, associated with a rarefying microvasculature, leading to impaired vascular function and a complex inter-individual response to treatment and revascularization interventions.

### 3.4. Intimal Hyperplasia: The Unmet Challenge of Post-Operative PAD Management

Bypass surgery and endovascular revascularization, which includes angioplasty, stenting and atherectomy, are recommended for patients with lifestyle-limiting claudication who do not respond to medical and/or exercise therapy. Unfortunately, the vascular trauma associated with surgical revascularization eventually leads to secondary occlusion of the injured vessel, a process called restenosis. For open surgical procedures such as bypass and endarterectomy, the rate of restenosis at 1-year ranges from 20 to 30% [62]. For endovascular approaches, the rate of re-occlusion after balloon angioplasty and stenting ranges from 30 to 60% depending on the location [63]. Restenosis has various causes, such as secondary growth of atherosclerotic lesions or inward remodeling. However, the most common cause is intimal hyperplasia (IH). IH is a well-known complication of all types of vascular surgery. The progressive growth of a neointimal layer causes both an outward and inward remodeling of the vessel wall, resulting in luminal narrowing and ultimately impaired perfusion of downstream organs.

IH begins as a physiological healing response to injury to the blood vessel wall [64,65]. Like atherosclerosis, IH is initiated by EC injury, which promotes vasoconstriction, platelet aggregation and recruitment/activation of resident and circulating inflammatory cells. Inflammation leads to the reprogramming of VSMC and fibroblasts into proliferating and migrating cells that form a neointimal layer between the intima and the internal elastic lamina. This new layer is mainly composed of VSMC-derived cells expressing various markers of mesenchymal (stemness) or osteochondrogenic phenotype and secreting abundant ECM [65,66,67].

All current strategies to limit IH, such as paclitaxel and sirolimus, target cell proliferation. Paclitaxel is a chemotherapeutic agent that stabilizes microtubules, thereby preventing mitosis [68]. Sirolimus inhibits the mammalian target of rapamycin (mTOR), a master regulator of cell growth and metabolism [66]. However, targeting cell proliferation to reduce IH also impairs re-endothelialization. Endothelial repair is critical to limit inflammation, remodeling and IH. Poor endothelial repair also prolongs the need for antithrombotic therapy.

The increasing number of PAD and CLTI patients in need of surgical vascular repair, combined with difficult long-term pharmacological and surgical management, calls for novel therapies to promote endothelial repair while inhibiting VSMC phenotypic switch, fibrosis, and VMC. The gaseous vasodilator molecule H_2_S has interesting properties in this respect.

## 4. Hydrogen Sulfide

H_2_S is a colorless, water-soluble, flammable, and highly toxic gas with a distinctive rotten-egg odor. In the last few years, H_2_S has been recognized as a novel gasotransmitter, not unlike NO and carbon monoxide [69].

Under physiological conditions (pH 7.4), H_2_S is mostly present as HS^−^. It acts as a reductant and undergoes a complex oxidation reaction to thiosulfate, sulfenic acids, per-sulfides, polysulfides and sulfate [70]. These oxidative products trigger post-translational modification of proteins by S-sulfhydration, also known as persulfidation, a chemical reaction that forms a persulfide group (R-SSH) on reactive cysteine residues [71]. For persulfidation to occur, cysteine residues or H_2_S must first be oxidized, for example in the form of polysulfides H_2_Sn. H_2_S and other forms of sulfide contribute to the homeostasis of numerous systems, including the cardiovascular, neuronal, gastrointestinal, respiratory, renal, hepatic, and reproductive systems [69]. A few high-throughput studies on the conversion of protein cysteinyl thiols (-SH) to persulfides (-SSH) showed extensive persulfidation of cysteine residues in response to H_2_S in different experimental designs [70,72,73,74,75,76].

### 4.1. Endogenous H_2_S Production

H_2_S is involved in many physiological and pathological processes [69]. In this section, we will introduce the biosynthesis of endogenous H_2_S and the regulation of H_2_S in mammalian tissues. 

Endogenous H_2_S production in mammals results from the oxidation of the sulfur-containing amino acids cysteine and homocysteine via the reverse ‘‘transsulfuration’’ pathway. H_2_S is produced by two pyridoxal 5′-phosphate (PLP)-dependent enzymes: cystathionine γ-lyase (CSE) and cystathionine β-synthase (CBS). CBS catalyzes the formation of cystathionine from homocysteine, which is subsequently converted to cysteine by CSE. Two other PLP-independent enzymes, 3-mercaptopyruvate sulphurtransferase (3-MST) and cysteine aminotransferase (CAT), generate sulfur, which is further processed to H_2_S. CAT converts L-cysteine to 3-mercaptopyruvate (3MP), which is converted to pyruvate and H_2_S by 3-MST in the presence of thioredoxin [77]. It should be noted that 3-MST mainly synthesizes H_2_S in the mitochondria (Figure 1).

Other mitochondrial enzymes such as persulfide dioxygenase (ETHE1), sulfide-quinone oxidoreductase (SQR), rhodanese (TST) and sulfite oxidase (SUOX) catalyze H_2_S oxidation to the metabolic end products sulfate and thiosulfate [78]. Moreover, cysteinyl-tRNA synthetase (CARS and CARS2) can synthesize CysSSH and cyshydropolysulfides (CysSnH), which can be further reduced to H_2_S [79] (Figure 1).

Although the enzymes and pathways responsible for endogenous H_2_S production are well understood, little is known about their relative contributions to circulating and tissular H_2_S and sulfane sulfur levels (e.g., polysulfides, persulfides, and thiosulfate). Accumulating evidence indicates that the enzymes involved in H_2_S production are often dysregulated in pathophysiologic conditions, leading to altered endogenous H_2_S production. All the evidence will not be listed here but we refer the reader to the extensive review by G. Cirino, C. Szabo and A. Papapetropoulos for a detailed account of the role, cellular distribution, and regulation of CSE, CBS, and 3-MST in mammalian tissues [69].

Briefly, the basal expression of *CBS* had been reported to be controlled by several transcription factors, including specificity protein (SP) 1 and 3, nuclear transcription factor-Y, and upstream transcription factor-1 (USF-1) [69]. CBS is mainly expressed in the central nervous system, the liver, and the pancreas, but is also found in most other systems, including the cardiovascular system. It has been mostly reported as a cytosolic enzyme, although CBS is also found in the mitochondria. CSE is a cytosolic and mitochondrial enzyme highly expressed in the liver and kidney. In the cardiovascular system, it is mainly expressed in EC [69]. In EC, *CSE* expression has been shown to be under the control of the activating transcription factor 4 (ATF4), which is selectively induced via the eukaryotic initiation factor 2 alpha (eIF2α) in response to various stresses such as ER-stress or amino acid restriction [80]. S. Bibli et al. recently demonstrated that CSE expression in EC is negatively regulated by shear stress, as opposed to eNOS in the mouse aorta [81]. This is in line with a previous study showing that only disturbed flow regions show discernable CSE protein expression after carotid artery ligation in the mouse [82]. Oxidative stress (H_2_O_2_) enhances cellular H_2_S production through the promotion of CSE activity [83]. 3-MST is expressed both in the mitochondria and cytosol, although most studies focus on the mitochondrial role of 3-MST [69]. 3-MST is found in most mammalian cells and tissues but varies between organs. *3-MST* is most abundantly expressed in the liver, kidney, testes, and brain, and *3-MST* expression is lowest in the spleen, thymus, lungs, and gut. Smoking, endurance exercise training, genetic defects and down syndrome have been reported to induce *3-MST* expression in various models [69].

Additional sources of H_2_S and related sulfur species also contribute to sulfur biology. In the gastrointestinal tract, anaerobic bacterial strains such as *E. coli*, *S. enterica*, *Clostridia* and *E. aerogenes* all convert cysteine to H_2_S, pyruvate and ammonia by means of cysteine desulfurases. These cysteine desulfurases are also involved in the formation of a protein-bound cysteine persulfide intermediate, which leads to the conversion of L-cysteine to L-alanine and sulfane sulfur [84].

In addition to this enzymatic production, there are several non-enzymatic pathways. Commensal bacteria use sulfite reductases to reduce sulfate or other organic oxidized sulfur compounds, resulting in the formation of H_2_S [85,86]. Several studies have associated these sulfate-reducing bacteria (SRB) with inflammation, inflammatory bowel syndrome and colorectal disease [87]. SRB colonize the intestines of ~50% of humans [86,87,88].

### 4.2. Vascular Properties of H_2_S and Benefits in the Context of Peripheral Arterial Disease (PAD and CLTI)

H_2_S participates in the homeostasis of many organs and systems. In the cardiovascular system, H_2_S mostly has beneficial effects, and protects against vascular diseases through several processes, including the attenuation of oxidative stress and inflammation, improving EC function and NO production and vasodilation, as well as the preservation of mitochondrial function [69]. *CSE* gene expression and CSE protein activity, as well as free circulating H_2_S, are reduced in human suffering from vascular occlusive diseases [89,90]. It was also recently demonstrated that, in patients undergoing vascular surgery, higher circulating H_2_S levels were associated with long-term survival [91], suggesting low H_2_S production as a risk factor for cardiovascular diseases. In the following sections, we will focus on the role of H_2_S in the vascular system and H_2_S properties relevant to vascular conditions.

#### 4.2.1. H_2_S Is a Potent Vasodilator

H_2_S is commonly known as a vasodilator [92]. One of the first reports came from Hosoki et al. in 1997, showing that H_2_S promoted NO-induced VSMC relaxation in rat thoracic aorta [93]. Then, numerous studies showed that H_2_S decreases blood pressure in spontaneously hypertensive rats (SHRs) [94,95,96] and salt-sensitive hypertension in Dahl rats [97].

Mechanistically, H_2_S triggers endothelium-independent vasorelaxation by persulfidation/activation the ATP-dependent potassium channel (K_ATP_) complex, specifically the regulatory sulfonylurea receptor subunit 1 and the pore-forming subunit Kir6.1 in VSMC (reviewed in [92,98]). Activation of the K_ATP_ channel and K^+^ export results in VSMC hyperpolarization and inhibition of voltage-dependent Ca^2+^ channels (VDCC), reduced [Ca^2+^]_i_ and relaxation [98]. H_2_S may also directly inhibit VDCC in VSMC [99,100]. In EC, H_2_S activates Ca^2+^ influx through TRPV4 [101], which in turn (i) increases eNOS expression/activity and NO production and VSMC vasodilation [102,103]; (ii) increases PLA2-mediated formation of arachidonic acid metabolites and VSMC relaxation; (iii) stimulates the large-conductance Ca^2+^-activated potassium channels (BK_Ca_), leading to EC hyperpolarization and subsequent hyperpolarization of adjacent VSMC, closure of VDCC and relaxation [98]. In VSMC, H_2_S may also enhance Ca^2+^ spark-induced large conductance potassium channel activation, facilitating VSMC relaxation [98]. Elevation in intracellular Ca^2+^ level in EC also leads to the activation of calmodulin, which in turn stimulates *CSE* expression to produce more H_2_S [99]. In addition, H_2_S promotes NO-dependent relaxation via enhanced eNOS activity due to persulfidation of Cys443 [70] (Figure 2).

Although H_2_S is usually described as a vasodilator gasotransmitter, recent studies demonstrated that H_2_S can also promote vasoconstriction. Thus, while concentrations of NaHS in the µM range induced vessels vasodilation [104], NaHS concentrations in the pico-nanoM range may stimulate contraction of VSMC [105] and rat coronary artery [106]. However, it should be noted that H_2_S alone does not trigger vasoconstriction, but only promotes constriction of precontracted vessels, enhancing the already existing tone. Enhanced vasoconstriction seems mediated by activation of Na^+^, K^+^, 2Cl^−^ cotransport and Ca^2+^ influx via VDCC [105]. H_2_S may also act via scavenging of NO [107]. This highlights the complexity of H_2_S contribution to the regulation of arterial blood pressure. Additionally, H_2_S may differentially act on the vascular tone depending on the arterial bed (carotid vs. mesenteric artery), the vessel type and size (conduit vs. resistant; capillary vs. larger vessels) (for full review, see [69,92,108]).

Overall, and although H_2_S is a potent vasodilator, very little is known about the role of CSE, CBS and 3MST-mediated H_2_S production in the regulation of blood pressure in physiologic and pathophysiologic conditions. Of note, the expression of H_2_S producing enzymes and substrate-dependent H_2_S production are decreased in humans with hypertension [109,110]. In addition, hypertensive patients with decreased endogenous H_2_S level have been shown to display microvascular endothelial dysfunction and impaired endothelium-dependent vasorelaxation [109]. Furthermore, the H_2_S precursor N-acetylcysteine decreased systolic and diastolic blood pressures in a clinical trial with 126 hypertensive patients [111]. It was also recently shown than a 6-week antihypertensive treatment with the sulfhydryl-donating angiotensin converting enzyme (ACE) inhibitor Captopril improved cutaneous microvascular endothelium-dependent vasodilation in middle-aged adults with hypertension [112]. This evidence indicates that H_2_S deficiency probably contributes to the development of hypertension and that H_2_S-based therapies may be of use for treatment of hypertension. 

#### 4.2.2. H_2_S Protects against Atherosclerosis

Atherosclerosis is a chronic progressive inflammatory disease. It is characterized by the accumulation of cholesterol-rich fatty deposits in the arterial tree. This disease involves numerous pathophysiological processes. These include EC dysfunction, vascular inflammation and lipoprotein accumulation, and VMSC proliferation and migration (see Section 3.1).

Impaired H_2_S production in *Cse*^−/−^ mice promotes atherosclerosis [113,114]. In contrast, the H_2_S donors NaHS [114,115,116] and GYY4137 [117] reduce the extent of vascular lesions in *ApoE*^−/−^ mice under high fat diet. S-aspirin (ACS14), a H_2_S-releasing form of aspirin, also protects *ApoE*^−/−^ mice against atherosclerosis [118].

H_2_S has been shown to protect against atherosclerosis mostly via anti-inflammatory (for full review, see [119]) and antioxidant effects (Figure 3). H_2_S possibly reduces inflammation mainly via inhibition of nuclear factor kappa B (NF-κB) [113,117,120,121]. NF-κB is a master regulator of pro-inflammatory genes, including cytokines and cell adhesion molecules. NaHS inhibits NF-κB activity via persulfidation/stabilization of Inhibitory kinase of NFκB (IκB) [122], which prevents NF-κB translocation to the nucleus [123]. In EC, inhibition of NF-κB leads to decreased expression of adhesion molecules vascular cell adhesion molecule 1 (VCAM-1) and intercellular adhesion molecule-1 (ICAM-1), thereby limiting recruitment of leukocyte to the aortic wall [113,117,121,124]. NF-κB inhibition also decreases production of pro-inflammatory cytokines and chemokines, including TNF-α, IL-1β, IL-6, and CCL2 [121,125,126]. In macrophages, H_2_S-mediated peroxisome proliferator activated receptor gamma (PPARγ) inhibition also inhibits C-X3-C chemokine fractalkine (CX3CL1) signaling in the context of atherosclerosis in *ApoE*^−/−^ mice [118]. H_2_S also inhibits TNF-α expression in EC in a model high glucose-induced vascular inflammation [127].

In addition, H_2_S was reported to inhibit leukocyte adherence to the endothelium via activation of ATP-sensitive K^+^ channels between EC and monocytes [128]. Moreover, S-sulfhydration of human antigen R (on Cys13) by CSE-derived H_2_S prevents its homodimerization and activity, which attenuates the expression of target proteins such as E-selectin and cathepsin S, which are linked to EC activation and atherosclerosis [74]. Exogenous H_2_S also promotes macrophage migration and shift toward the M2, pro-resolution phenotype [129,130,131]. However, further studies are required to identify whether H_2_S has a direct effect on macrophage state. Moreover, the fact that H_2_S stimulates eNOS activity and NO production in EC has been shown to contribute to its anti-inflammatory effect in the context of atherosclerosis [113,132]. The anti-inflammatory property of H_2_S may also involve inhibition of cyclooxygenase (COX2) expression and secretion of prostaglandin PGE2, which stimulates the secretion of pro-inflammatory cytokines and monocyte adhesion to EC [133]. H_2_S has also been proposed to protect EC from inflammation by inhibiting the NOD-, LRR- and pyrin domain-containing protein 3 (NLRP3) inflammasome in atherosclerotic conditions [134].

H_2_S also protects against atherosclerosis via antioxidant effects (Figure 3). Excessive production of reactive oxygen species (ROS), such as superoxide anions O_2_^−^, H_2_O_2,_ and NO, leads to cellular and molecular damages. Oxidative stress is linked to the inflammatory process and contributes to the progression of PAD [135]. H_2_S is an antioxidant that can directly reduce ROS. Thus, NaHS protects myocytes and contractile activity by scavenging oxygen-free radical (O_2_^−^, H_2_O_2_), thereby decreasing lipid peroxidation [136]. In the context of atherosclerosis, NaHS was shown to reduce O_2_^−^ formation [115]. H_2_S also prevents LDL oxidation and formation of oxidized LDL particles (ox-LDL), resulting in reduced foam cell formation [137,138]. Interestingly, ox-LDL triggers the hypermethylation of the CSE promoter, thus decreasing CSE expression and H_2_S production in murine macrophages [121,139]. Mitochondrial respiration is a major source of ROS [140,141] and H_2_S binds the copper center of cytochrome c oxidase (complex IV), thereby inhibiting respiration and limiting ROS production [142].

H_2_S also upregulates antioxidant defenses, in particular the nuclear factor erythroid 2-related factor 2 (NRF2) pathway (reviewed in [143]) (Figure 3). NRF2 is a major transcription factor that regulates antioxidant genes including heme oxygenase 1 (HO-1), thioredoxin-1 (TRX-1) and glutathione peroxidase (GPx). H_2_S promotes NRF2 activity via persulfidation of Keap-1 on Cys131, leading to dissociation of the cytosolic KEAP1-NRF2 complex, and nuclear translocation of NRF2 to induce the expression of its target genes. Thus, the H_2_S donor GYY4137 mitigates diabetes-accelerated atherosclerosis via improved Nrf2 activation in *Ldlr*^−/−^ mice, which induces *Ho-1* expression and reduces superoxide formation [144]. Exogenous H_2_S might protect arterial EC through antioxidant proprieties by activating the NRF2 pathway [145]. H_2_S also increases glutathione (GSH) production via modulation of the transulfuration pathway. GSH is an antioxidant that protects cells by reducing ROS. H_2_S interaction with GSH has been studied in detail in the central nervous system, where GSH plays a major role in maintaining the homeostasis between antioxidant and ROS production (reviewed in detail in [146]). In the vascular system, H_2_S persulfidates the GPx1, which promotes GSH synthesis and results in decreased lipid peroxidation in the aortic wall in the context of atherosclerosis [147]. H_2_S also stimulates *TRX-1* expression, via silencing the expression of inhibitory protein Trx-interacting protein (*TXNIP*) [148,149,150,151]. Trx-1 is instrumental in the cardioprotective effects of H_2_S against ischemia-induced heart failure [150]. Trx-1 has atheroprotective effects via suppression of NLRP3 expression in macrophages after ox-LDL stimulation [152]. Trx-1 also promotes the M2 pro-resolutive macrophages state in *ApoE*^−/−^ mice [153]. TRX-1 also suppresses Nox4 activity and ROS production in HUVEC exposed to ox-LDL [154].

H_2_S biosynthesis also occurs in adipocytes. Increased adiposity-enhanced oxidative stress and obesity-related low grade adipose tissue inflammation play a crucial role in the development of atherosclerosis [155]. The perivascular adipose tissue (PVAT), in particular, has been proposed to contribute to cardiovascular pathogenesis by promoting ROS generation and inflammation. The PVAT is the fourth outer layer of vessels surrounding the vasculature, which has emerged as an active modulator of vascular homeostasis and pathogenesis of cardiovascular diseases [156,157]. The adipose tissue is a very active endocrine tissue, secreting a variety of adipokines, including leptin and adiponectin, and pro- inflammatory cytokines such as TNFα IL-1β and IL-6. Leptin has been found to promote atherosclerosis, whereas adiponectin has been shown to have anti-inflammatory and anti-atherogenic effects [158,159,160]. H_2_S could reduce atherosclerosis by the inhibition of adipogenesis [161]. H_2_S deficiency may affect the process of adipocyte maturation and lipid accumulation. *3-MST* knockdown also facilitated adipocytic differentiation and lipid uptake. The 3-MST/H_2_S system plays a tonic role in suppressing lipid accumulation and limiting the differentiation of adipocytes [162].

Overall, H_2_S has been found to be cytoprotective in oxidative stress in a wide range of physiologic and pathologic conditions.

#### 4.2.3. H_2_S Protects against Vascular Medial Calcification

First and foremost, H_2_S can protect from arterial calcification indirectly. As stated in Section 3.2, chronic kidney disease and diabetes mellitus are the leading causes of VMC. H_2_S has been shown to provide benefits against both pathologies. These will be not discussed in this review due to space constraints. Readers interested in a more in-depth analysis of the benefits of H_2_S against diabetes are referred to other reviews [163,164]. H_2_S has been shown to decrease blood glucose, atherosclerosis, and diabetic cardiomyopathy in the context of diabetes in pre-clinical models [165,166]. H_2_S also provides renal protection against various injury, including models of diabetic nephropathy [167]. Below we detail the studies directly measuring the impact of H_2_S on the process of VMC in various experimental in vitro and in vivo models (Figure 4).

VMC is an accumulation of Ca^2+^ and inorganic phosphate (Pi) in arteries with mineral deposits in the intimal or medial layer of the vessel wall [168,169]. VMC formation is a complex, controlled molecular process involving the differentiation of macrophages and VSMC into osteoclast-like cells, like that which occurs in bone formation [170,171] (see Section 3.2). In recent years, H_2_S supplementation has been shown to lessen VMC. In this section, we discussed these studies and their molecular insight into the potential mechanisms underlying the benefits of H_2_S on VMC (Figure 4).

Using a model of VMC by administration of vitamin D3 plus nicotine (VDN), it was shown in rats that *Cse* expression is downregulated in the context of VMC, and that treatment with H_2_S donors NaHS [172] or AP39 [173] lessens VMC in that model. Similarly, exogenous NaHS treatment also restored Cse activity and expression, and inhibited aortic osteogenic transformation in a rat model of diabetic nephropathy [163]. NaHS also limits Ca^2+^ deposition in VSMC in in vitro models of calcification in cell culture [138,174,175]. 

Mechanistically, H_2_S has been proposed to limit VMC via reduced ER stress-induced VSMC phenotypic reprogramming [173]. H_2_S attenuates VSMC calcification induced by high levels of glucose and phosphate through upregulating elastin level via the inhibition of the signal transducer and activator of transcription 3 (Stat3), leading to reduced Cathepsin S expression [175]. NaHS also significantly reduced Stat3 activation, cathepsin S activity in a rat model of diabetic nephropathy [163]. In another model of VSMC calcification induced by circulating calciprotein particles, H_2_S was shown to mitigate VMC via activation of the antioxidant factor NRF2 [174]. Overall, H_2_S likely acts on several pathways improving VSMC identity to avert osteogenic transformation (Figure 4). Of note, low plasma levels of H_2_S and decreased CSE enzyme activity were found in patients with chronic kidney disease receiving hemodialysis [138,176], suggesting that low H_2_S may contribute to VMC in patients. 

From a translational point of view, it should be mentioned that the FDA-approved H_2_S donor Sodium thiosulfate (STS) reduces periarticular calcification in a mouse model of osteoarthritis via its effects on chondrocyte mineralization [177]. STS is already used in the clinic to treat cyanide poisoning and to increase the solubility of Ca^2+^ for the treatment of acute calciphylaxis, a rare vascular complication of patients with end-stage renal disease [178]. The phase III CALISTA trial of STS for acute calciphylaxis is ongoing (NCT03150420) and STS is also tested in a few clinical trials for the treatment of ectopic calcification (NCT03639779; NCT04251832; NCT02538939). Although STS has not been shown to reduce VC, it stands to reason that STS should be explored for the treatment of VMC in the context of PAD.

#### 4.2.4. H_2_S Supports Endothelial Cell Function

With one simple monolayer, the endothelium regulates vascular tone, cell adhesion and vessel wall inflammation, and VSMC phenotype. Atherosclerosis and PAD preferentially develop at site of disturbed arterial flow leading to “endothelium activation”. As described in the previous sections, impaired EC-derived H_2_S contributes to inflammation and oxidative stress, leading to atherosclerosis. The ability of EC to proliferate and migrate to restore the endothelial barrier of the vessel is a key feature in wound healing, vascular repair, and the resolution of inflammation. In this section, we describe the effects of H_2_S in EC proliferation and migration, which constitute an interesting avenue of research to promote therapeutical angiogenesis for PAD patients (Figure 5). The benefits of H_2_S on EC may also limit MVD, which contributes to the severity of PAD (see Section 3.3).

Preclinical studies have shown that H_2_S and polysulfites stimulate EC angiogenesis and arteriogenesis. Thus, H_2_S donors stimulate the growth, motility, and organization of EC into a vascular structure in vitro [179]. Conversely, inhibition of H_2_S biosynthesis, either by pharmacological inhibitors or by silencing CSE, CBS or 3MST, reduces EC growth and migration in vitro [180,181]. *Cse*^−/−^ mice also show reduced vascular endothelial growth factor (VEGF)-induced sprouting angiogenesis in the mouse aortic ring assay ex vivo [182]. In vivo studies on chicken chorioallantoic membranes treated with the CSE inhibitor propargylglycine (PAG) also indicate that CSE is important for vascular branching [182]. In vivo, there is no adequate PAD model. Most studies are conducted using the hindlimb ischemia (HLI) model, which can be applied to rodent and pigs alike. In this model, transection or occlusion of the femoral or iliac artery leads to ALI. Recovery from ALI is then followed for 2 to 4 weeks via angiographic scores, return of hind limb blood flow, and capillary density in the gastrocnemius muscle. As such, the model allows for assessment of arteriogenesis and angiogenesis-mediated neovascularization. Using this model, it was shown that whole-body *Cse*^−/−^ mice with impaired H_2_S production displayed impaired neovascularization [114,183]. Conversely, we recently showed that *Cse* overexpression in transgenic mice is sufficient to promote neovascularization following HLI [184]. Various H_2_S donors such as NaHS, GYY4137, ZYZ-803, a hybrid NO and H_2_S donor were also shown to improve capillary density, angiographic scores, and hind limb blood flow in rodent models [179,185,186]. Fu et al. also reported that H_2_S-saturated water accelerates perfusion recovery through improved arteriogenesis in the abductor muscle and increased capillary density in the gastrocnemius muscle in the mouse [187]. Diallyl trisulfide, S-allylcysteine and S-propyl-L-cysteine, organosulfur compounds found in garlic, were also shown to improve blood flow recovery after HLI in mice in various context [188,189,190,191,192,193]. Rushing et al. also showed that SG1002, a H_2_S-releasing pro-drug, increases leg revascularization and collateral vessel number after occlusion of the external iliac artery in the minipig [194]. We also recently showed that the H_2_S donor STS promotes EC proliferation and migration in vitro, and VEGF-induced angiogenesis in vivo. STS also accelerates neovascularization in the HLI model in WT and *Ldlr*^−/−^ male mice [195].

Several mechanisms have been proposed to explain H_2_S-induced angiogenesis (Figure 5). Most studies report that H_2_S promotes VEGF-driven sprouting angiogenesis. Thus, overexpression of CSE, CBS and 3-MST leads to an increase in VEGF expression and decrease in anti-angiogenic factor endostatin [196]. Similarly, NaHS increases VEGF expression while reducing the levels of anti-angiogenic factors [197]. In EC, H_2_S induces the VEGF receptor VEGFR2 persulfidation, which facilitates dimerization, autophosphorylation and activation [198]. Interestingly, short-term exposure of human EC to VEGF increases H_2_S production [182], suggesting a positive feedback loop of VEGF signaling through H_2_S. Matrigel plug angiogenesis assay also confirmed the importance of CSE and H_2_S in VEGF-induced angiogenesis [195,199,200]. *CSE* overexpression is also sufficient to stimulate VEGF-dependent EC migration in vitro, and capillary formation using an aortic ring assay ex vivo [184]. CSE and H_2_S are also required for VEGF-dependent EC migration and angiogenesis in response to amino acid restriction [80]. Exogenous H_2_S donors have also been shown to stimulate the growth pathways Akt, p38 and ERK1/2, which all promote EC proliferation and migration [182,200,201]. EC migration is also activated by exogenous H_2_S through K_ATP_ channels/MAPK pathways in vitro [182]. CSE overexpression has also been reported to increase cGMP level [199], which fuels capillary tube formation [187]. In addition, H_2_S promotes angiogenesis via interactions with NO, which is essential for EC survival and growth during VEGF- or bFGF-induced angiogenesis [202]). Finally, H_2_S is proposed to promote angiogenesis by inhibiting mitochondrial electron transport and oxidative phosphorylation, increasing glucose uptake and glycolytic ATP production required to rapidly power EC migration [80]. Indeed, under hypoxia when mitochondrial respiration is not possible, glycolysis fuels EC migration and proliferation during angiogenesis [203,204]. H_2_S promotes the metabolic switch in EC to favor glycolysis, which drives VEGF-induced EC migration [80,205] (Figure 5).

#### 4.2.5. H_2_S Inhibits Intimal Hyperplasia: Post-Operative Management of CLTI Patients

The revascularization procedure in CLTI patients is plagued by restenosis of the operated area, a progressive reduction of the vessel lumen at the site of angioplasty, or at the anastomosis of a bypass graft. Restenosis is mainly related to a complex phenomenon called IH (see Section 3.4). IH is characterized by a thickened wall due to VSMC proliferation and deposition of a proteoglycan-rich ECM between the endothelium and the internal elastic lamina.

Mice lacking *Cse* show a significant increase in IH formation as compared to WT mice in a model of carotid artery ligation [205,206]. On the contrary, *Cse* overexpression decreases IH formation in a murine model of vein graft by carotid-interposition cuff technique [207]. We and others demonstrated that systemic treatment using diverse H_2_S donors inhibit IH in vivo in various models in rats [208], rabbits [209] and mice [205,206,210]. We also showed that various H_2_S donors inhibit IH ex vivo in a model of vein graft IH [205,210,211]. Recently, it was shown that a locally applicable gel containing the hydrogen sulfide releasing prodrug (GYY4137) mitigates graft failure and improves arterial remodeling in a model of vein graft surgery in the mouse [212]. We also recently showed that a H_2_S-releasing biodegradable hydrogel inhibited VSMC proliferation while facilitating EC proliferation and migration, which inhibited IH in an ex vivo model of human vein graft disease [211].

H_2_S probably reduces IH mainly via inhibition of VSMC proliferation (Figure 6). Indeed, several studies demonstrated that H_2_S supplementation using various donors, or CSE overexpression, decreases VSMC proliferation [205,209,210,211,213]. H_2_S also specifically inhibits VSMC migration. Thus, Several H_2_S donors have also been shown to reduce VSMC migration in vitro [205,210,211]. VSMC isolated from *Cse*^−/−^ mice also migrate faster than wild type VSMC, and blocking CSE activity using PAG increases VSMC migration [206,214].

The mechanisms whereby H_2_S affects VSMC proliferation and migration are not fully understood (Figure 6). In mouse VSMC, H_2_S has been shown to modulate the MAPK pathway, especially ERK1,2 [208], and Ca^2+^-sensing receptors [215,216]. H_2_S may also limit MMP2 expression and ECM degradation, preventing VSMC migration from the media to the intima [206,214]. In human VSMC, we reported that the H_2_S donor Zofenopril decreases the activity of the MAPK and mTOR pathways, which correlates with reduced VSMC proliferation and migration [210]. We also showed that the H_2_S donors NaHS and Sodium thiosulfate (STS; Na_2_S_2_O_3_) inhibit microtubule polymerization, which results in cell cycle arrest and inhibition of proliferation and migration in primary human VSMC [205]. Interestingly, an ongoing clinical study aims to evaluate the efficacy and safety of STS compared to placebo on myocardial infarct size in ST-segment elevation myocardial infarction (STEMI) patients treated with percutaneous coronary intervention (NCT02899364). The anti-inflammatory properties of H_2_S may also contribute to reduced IH [217,218] and it was recently shown that NaHS prevents IH through activation of the Nrf2/HIF1α pathway [219].

### 4.3. Further Directions and Limitations

Although H_2_S research is still in its early stages, there is considerable evidence to suggest that this gas plays a protective role in the development of cardiovascular disease. As mentioned throughout the review, H_2_S acts in concert with NO, and the vascular effects of NO and H_2_S are mutually supportive and intertwined (for a complete review, see [69]). Due to poor tolerability and uncontrolled hypotensive effects, all therapeutic strategies based on NO have failed. Whether H_2_S-based solutions can succeed where NO has failed remains to be seen. There is currently no clinically approved molecule that exploits the therapeutic potential of H_2_S. Most compounds available for research have poor translational potential due to their pharmacokinetic properties. Developing stable H_2_S donors that allow slow and sustained H_2_S release over months/years will be the first challenge. Given the instability and short half-life of H_2_S, such molecules are difficult to design. Another challenge for systemic or local H_2_S release is the delivery system, as H_2_S donors may require a carrier system. Gels, nanoparticles, multilayer coatings, and biodegradable scaffolds were invented for sustained release. Applying this knowledge to H_2_S donors will be interesting. Another strategy to harness the benefits of H_2_S is to conjugate the H_2_S-releasing moiety with well-established parent compounds. For example, the sulphhydrylated ACEi zofenopril has been shown to improve clinical outcomes in patients with various cardiovascular diseases such as acute myocardial infarction and congestive heart failure [220,221,222]. S-aspirin (ACS14), an H_2_S-releasing form of aspirin, and otenaproxesul, an H_2_S-releasing non-steroidal anti-inflammatory drug developed by Antibe Therapeutics Inc, may also prove beneficial for vascular patients. Further work is needed to evaluate the therapeutic potential of these molecules against atherosclerosis, but also against VMC and MVD in PAD. H_2_S-eluting balloons and stents would be interesting tools to limit VSMC proliferation while promoting EC recovery to limit IH in PAD/CLTI patients requiring surgery.

Strategies to increase endogenous H_2_S production using small molecules or diet are also explored. However, further animal studies are needed to understand and leverage endogenous H_2_S production and to test the potential and safety of new H_2_S-based therapies.

## 5. Conclusions

PAD is a chronic, recurrent disease with a major impact on quality of life and devastating long-term clinical outcomes. PAD remains underdiagnosed and undertreated compared to other atherosclerotic diseases such as myocardial infarction and stroke. Emerging evidence suggests that PAD has different pathological features in peripheral vessels compared to the well-characterized coronary arteries, in particular media calcification and microvascular dysfunction. In addition, the incidence of restenosis following surgical revascularization remains high. The increasing number of PAD and CLTI patients, combined with difficult long-term pharmacological and surgical management, warrants further research to better understand the molecular mechanisms of PAD.

Although still in its early stages, research into H_2_S suggests its potential to protect against cardiovascular disease. The success of H_2_S-based solutions remains uncertain and there are currently no clinically approved molecules exploiting its therapeutic potential. The development of stable H_2_S donor molecules for sustained release is challenging due to the instability of the gas. Delivery systems such as gels, nanoparticles and biodegradable scaffolds designed for sustained release could be applied to H_2_S donors. H_2_S-eluting balloons and stents may be useful in limiting VSMC proliferation and promoting EC recovery in patients with PAD. Another strategy is to combine H_2_S donors with established drugs. Strategies to increase endogenous H_2_S production using small molecules or diet are also investigated. Further studies are needed to explore the therapeutic potential and safety of these molecules against atherosclerosis, vascular calcification, and microvascular dysfunction. The advancement of this knowledge will contribute to the development of successful H_2_S-based therapies in the future.

## Data Availability

Not applicable.

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
