# Peer review of "Clinical Potential of Hydrogen Sulfide in Peripheral Arterial Disease"

_ijms, 2023, doi:10.3390/ijms24129955_

Round 1
Reviewer 1 Report
Dear Author,
Your review on the role of H2S in atherosclerosis is well-written and described in an easy-to-read way.
I would only suggest adding information on the role of interleukins and eicosanoids (especially leukotrienes) in the introductory sections.
Minor suggestions:
1) CLTI stands for "chronic limb-threatening ischaemia", not critical limb threatening ischaemia. Please correct all along the manuscript.
2) Lines 48-49: "Venous bypass surgery and endovascular approaches such as angioplasty and stenting are the main treatment for CLTI." - you missed endarterectomy.
3) Lines 61-91 - this section needs to be reorganized as in present form is really chaotic. Divide it into 3 subsections: CLTI (Rutherford 4-6), severe IC (RTH 3), and mild PAD (RTH 1-2). Treatment depends on the severeness of PAD and differs a lot between those groups.
4) Lines 70-72: "Whenever possible, patients with PAD should also receive antithrombotic agents as these significantly reduce the risk of stroke, myocardial infarction, and ALI." - This statement is against the present guidelines. Patients with symptomatic PAD should be always treated with at least single antiplatelet therapy and if possible combined with Rivaroxaban 2x2,5mg. There are no guidelines on the preventive use of other anticoagulants. The references given are just single studies, not guidelines.
5) Line 85: Approximately 25% of CLTI patients are ineligible for revascularisation 85 and amputation is often the only option [22]." - only 1 study gives such a high value, please change to "up to" instead of "approximately".
6) Lines 122-125: "PAD is commonly described as an atherosclerotic disease. However, for lower limb artery disease, recent clinical data suggest that we have overestimated the contribution of atherosclerosis and underestimated the role of medial arterial calcification in PAD (recently reviewed in detail in [30, 31])." - Calcifications are strictly related to atherosclerosis. I would rather say "However, for lower limb artery disease, recent clinical data suggest that we underestimated the role of medial arterial calcification in PAD (recently reviewed in detail in [30, 31])."
7) Please check if the newest references are used. In some cases, the dates of publication are quite old.
Kind regards
Minor spelling changes need to be done.
Author Response
Below is our detailed answers to the comments raised by Reviewer 1
Dear Author,
Your review on the role of H2S in atherosclerosis is well-written and described in an easy-to-read way. I would only suggest adding information on the role of interleukins and eicosanoids (especially leukotrienes) in the introductory sections.
We thank this reviewer for his positive feedback and useful comments. We now give more details on the role of interleukins and eicosanoids in the introduction. We agree with the reviewers that section 3.1. Atherosclerosis was a bit too brief. However, the aim of this review was not to give an exhaustive overview of the pathogenesis of atherosclerosis. New references have been included, which summarize the current knowledge on cellular participants and key inflammatory signalling pathways in atherosclerosis.
Minor suggestions:
1) CLTI stands for "chronic limb-threatening ischaemia", not critical limb threatening ischaemia. Please correct all along the manuscript.
Thank for the correction. It has been corrected throughout the manuscript.
2) Lines 48-49: "Venous bypass surgery and endovascular approaches such as angioplasty and stenting are the main treatment for CLTI." - you missed endarterectomy.
Thank for the suggestion. Endarterectomy has been added.
3) Lines 61-91 - this section needs to be reorganized as in present form is chaotic. Divide it into 3 subsections: CLTI (Rutherford 4-6), severe IC (RTH 3), and mild PAD (RTH 1-2). Treatment depends on the severeness of PAD and differs a lot between those groups.
We agree with the reviewer that section 2. Current management of PAD and CLTI was poorly structured. The aim of this part was to provide an overview and not a complete list of recommendation. That said, this section has been restructured with the addition of the Fontaine and Rutherford stages.
4) Lines 70-72: "Whenever possible, patients with PAD should also receive antithrombotic agents as these significantly reduce the risk of stroke, myocardial infarction, and ALI." - This statement is against the present guidelines. Patients with symptomatic PAD should be always treated with at least single antiplatelet therapy and if possible combined with Rivaroxaban 2x2,5mg. There are no guidelines on the preventive use of other anticoagulants. The references given are just single studies, not guidelines.
We now include a more comprehensive description of the main recommendations based on the current guidelines as suggested.
5) Line 85: Approximately 25% of CLTI patients are ineligible for revascularisation 85 and amputation is often the only option [22]." - only 1 study gives such a high value, please change to "up to" instead of "approximately".
Modified as requested
6) Lines 122-125: "PAD is commonly described as an atherosclerotic disease. However, for lower limb artery disease, recent clinical data suggest that we have overestimated the contribution of atherosclerosis and underestimated the role of medial arterial calcification in PAD (recently reviewed in detail in [30, 31])." - Calcifications are strictly related to atherosclerosis. I would rather say "However, for lower limb artery disease, recent clinical data suggest that we underestimated the role of medial arterial calcification in PAD (recently reviewed in detail in [30, 31])."
Modified as requested
7) Please check if the newest references are used. In some cases, the dates of publication are quite old.
Recent references have replaced older references whenever possible. Only 21 out of 220 references are before 2010.
Reviewer 2 Report
Peripheral artery disease (PAD) affects over 230 million individuals globally, causing decreased quality of life and heightened risks for vascular complications and mortality. Despite its prevalence and negative consequences, PAD is often underdiagnosed and undertreated in comparison to myocardial infarction and stroke. The disease is caused by a combination of macrovascular atherosclerosis and calcification along with microvascular rarefaction, leading to chronic peripheral ischaemia. Due to its challenging long-term pharmacological and surgical management, novel therapies are required to address the increasing incidence of PAD. Hydrogen sulphide (H2S), a gasotransmitter derived from cysteine, has noteworthy properties such as vasorelaxation, cytoprotection, anti-inflammatory, and antioxidant effects. In this review, the authors discussed the current understanding of PAD pathophysiology, as well as the impressive advantages of H2S in combating atherosclerosis, inflammation, vascular calcification, and other vasculo-protective effects. It is an interesting review article, specific comments:
1. The article provides a detailed overview of PAD and its impact on patients’ quality of life, and the authors discuss the potential benefits of H2S in treating PAD due to its vasorelaxant, cytoprotective, antioxidant, and anti-inflammatory properties. The iconographies are particularly welcome for the review article to attract the readers. However, the quality (resolution) of some figures is not good enough for publication.
2. How does H2S work at a molecular level to produce these beneficial effects?
3. The article mentions that PAD is underdiagnosed and undertreated compared to other conditions such as myocardial infarction and stroke. Why is this the case?
4. The article mentions that current management strategies for PAD include lifestyle modification, pharmacotherapy, and surgical revascularization. How effective are these strategies in managing the disease?
5. The article discusses the role of medial arterial calcification in PAD, particularly in lower limb arteries. How does this differ from intimal calcification?
6. The authors mention that H2S can have both vasorelaxation and vasoconstriction effects depending on its concentration. How does this impact its potential use as a treatment for PAD?
7. Are there any potential side-effects of H2S?
8. In addition to H2S releasing pro-drugs, are there any drugs can regulate H2S production in the cells?
9. The article provides a detailed overview of the different enzymes involved in endogenous H2S production. How is H2S production regulated in mammalian tissues?
10. What are the limitations of current studies and what further research is needed in this area?
Author Response
Below is our detailed answers to the issues raised by Reviewer 2 (answers in blue)
In this review, the authors discussed the current understanding of PAD pathophysiology, as well as the impressive advantages of H2S in combating atherosclerosis, inflammation, vascular calcification, and other vasculoprotective effects. It is an interesting review article.
We thank the reviewer for his positive comment
specific comments:
The article provides a detailed overview of PAD and its impact on patients’ quality of life, and the authors discuss the potential benefits of H2S in treating PAD due to its vasorelaxant, cytoprotective, antioxidant, and anti-inflammatory properties. The iconographies are particularly welcome for the review article to attract the readers. However, the quality (resolution) of some figures is not good enough for publication.
All the figures have been remade in illustrator for final production.
- How does H2S work at a molecular level to produce these beneficial effects?
As stated throughout the second part of the review, H2S works on many levels to produce these beneficial effects via anti-inflammatory and antioxidant effects.
- The article mentions that PAD is underdiagnosed and undertreated compared to other conditions such as myocardial infarction and stroke. Why is this the case?
PAD is underdiagnosed and undertreated because its symptoms are more diverse, diffuse, and not immediately life-threatening. As stated in the introduction (line 35-36), intermittent claudication, the cardinal symptom of PAD, may be present in only 10-35% of patients, whereas 40-50% of PAD patients have a wide range of atypical leg symptoms, and 20-50% of patients are asymptomatic. Therefore, PAD diagnostic is more complicated and generally underdiagnosed and undertreated. Patients also tend to ignore the symptoms as they are not as severe as cardiac or stroke symptoms.
- The article mentions that current management strategies for PAD include lifestyle modification, pharmacotherapy, and surgical revascularization. How effective are these strategies in managing the disease?
This part has been modified as per the recommendations of reviewer 1 to better explain the management of PAD depending on severity. New references have been added.
- The article discusses the role of medial arterial calcification in PAD, particularly in lower limb arteries. How does this differ from intimal calcification?
To answer this question, I refer the reviewer to the excellent reviews included in the article, especially table 1 of references 31 (Kim TI, Guzman RJ: Medial artery calcification in peripheral artery disease. Front Cardiovasc Med 2023, 10:1093355. DOI: 10.3389/fcvm.2023.1093355.). New references have also been added describing in more detail medial calcification. Schemes 3 and 4 have been updated to better describe vascular calcification.
Intimal calcification is a feature of advanced atherosclerosis plaques. Intimal calcification occurs on the intimal side of the necrotic core of atherosclerotic plaques or along the edge of the fibrous cap and thought to contribute to plaque rupture.
Medial calcification is localized in the media layer. It is common in elderly individuals, as well as patients with diabetes, CKD, and hypertension, but mostly in smaller below the knee vessel, independently of atherosclerosis plaque. Unlike intimal calcification, medial calcification is not associated with the presence of lipid deposits, macrophages, and foam cells.
Both intimal and medial vascular calcification are highly regulated cell-mediated pathologies driven by pathways contributing to bone formation and osteogenic differentiation of stem cells, alterations in cell-matrix interactions, and phenotypic reprogramming of VSMC into osteogenic-like cells. However, medial calcification has been less studied and is far less understood and only starts to emerge a major contributor to lower limb PAD. Given that the histologic presentation and environment are largely different, it is likely that the molecular processes leading to medial vascular calcification are partly distinct from that of intimal calcification.
- The authors mention that H2S can have both vasorelaxation and vasoconstriction effects depending on its concentration. How does this impact its potential use as a treatment for PAD?
This part of the review was confusing as H2S alone does not trigger vasoconstriction. It has been modified as follows for clarity.
Start line 434: While concentrations of NaHS in the µM range induced vessels vasodilation, NaHS concentrations in the pi-co-nanoM range may stimulate contraction of VSMC. However, it should be noted that H2S alone does not trigger vasoconstriction, but only promotes constriction of precontracted vessels, enhancing the already existing tone. Enhanced vasoconstriction seems mediated by activation of Na+,K+,2Cl- cotransport and Ca2+ influx via VDCC. H2S may also act via scavenging of NO.
- Are there any potential side-effects of H2S?
As with any potent molecule there are potential side effects. Like NO donors, some H2S donors such as sodium thiosulfate have been reported to induce episodes of hypotension at injection. Larger dose can be lethal as H2S is a very potent inhibitor of cellular respiration.
- In addition to H2S releasing pro-drugs, are there any drugs can regulate H2S production in the cells?
There is nothing FDA approved to my knowledge. Small molecules screenings have been performed to find existing modulator of endogenous production. However, no study reported such molecules yet. The startup Sulficon (https://www.sulfiscon.ch/) is supposedly developing CSE allosteric modulators to boost endogenous H2S production in the cells. Overall, there is enough evidence to include it in the review.
- The article provides a detailed overview of the different enzymes involved in endogenous H2S production. How is H2S production regulated in mammalian tissues?
Little is known about the regulation of H2S production in mammalian tissues in healthy and pathological conditions. Section 4.1. Endogenous H2S production has been developed to describe regulation of H2S-producing enzymes. We refer the reader to the extensive review by G. Cirino et al. for a detailed account of the role, cellular distribution, and regulation of CSE, CBS, and 3-MST in mammalian tissues (Cirino G, Szabo C, Papapetropoulos A. Physiological Roles of Hydrogen Sulfide in Mammalian Cells, Tissues and Organs. Physiol Rev (2022). Epub 2022/04/19. doi: 10.1152/physrev.00028.2021.).
- What are the limitations of current studies and what further research is needed in this area?
A new section 4.3Further directions and limitations has been added to better described the road ahead and main challenges we face with H2S-based solutions.
Round 2
Reviewer 2 Report
Peripheral artery disease (PAD) affects over 230 million individuals globally, causing decreased quality of life and heightened risks for vascular complications and mortality. Despite its prevalence and negative consequences, PAD is often underdiagnosed and undertreated in comparison to myocardial infarction and stroke. The disease is caused by a combination of macrovascular atherosclerosis and calcification along with microvascular rarefaction, leading to chronic peripheral ischaemia. Due to its challenging long-term pharmacological and surgical management, novel therapies are required to address the increasing incidence of PAD. Hydrogen sulphide (H2S), a gasotransmitter derived from cysteine, has noteworthy properties such as vasorelaxation, cytoprotection, anti-inflammatory, and antioxidant effects. In this review, the authors discussed the current understanding of PAD pathophysiology, as well as the impressive advantages of H2S in combating atherosclerosis, inflammation, vascular calcification, and other vasculo-protective effects. The revision of the manuscript is much improved, no additional comments.